# Senescence and SASP Are Potential Therapeutic Targets for Ischemic Stroke

**DOI:** 10.3390/ph17030312

**Published:** 2024-02-28

**Authors:** Blake Ouvrier, Saifudeen Ismael, Gregory Jaye Bix

**Affiliations:** 1Department of Neurosurgery, Clinical Neuroscience Research Center, School of Medicine, Tulane University, New Orleans, LA 70112, USA; bouvrier@tulane.edu (B.O.); sismael@tulane.edu (S.I.); 2Tulane Brain Institute, Tulane University, New Orleans, LA 70112, USA; 3Department of Neurology, School of Medicine, Tulane University, New Orleans, LA 70112, USA; 4School of Public Health and Tropical Medicine, Tulane University, New Orleans, LA 70122, USA

**Keywords:** senescence, senescence-associated secretory phenotype, senolytics, neuroinflammation

## Abstract

Aging is a known co-morbidity of ischemic stroke with its risk and severity increasing every year past 55+. While many of the current stroke therapies have shown success in reducing mortality, post-stroke morbidity has not seen the same substantial reduction. Recently, the involvement of cellular senescence and SASP in brain injury and neurological degeneration has been recognized. Ischemic injury causes oxidative stress and mitochondrial damage that induces senescence through the activation of p21 and p16 pathways, ultimately leading to synthesis and release of senescence-associated secretory phenotype (SASP). This ischemic event causes stress-induced premature senescence (SIPS), aging the brain decades beyond the standard biological age due to an increase in senescent cells in the ischemic core and ipsilateral hemisphere. Therefore, therapies that target the senescent cells and SASP, including senolytics, senomorphic drugs, stem cell therapies, and other cell-specific interventions, may be a new path for stroke treatment.

## 1. Introduction of Cellular Senescence

Cellular senescence occurs when cells exit the cell cycle, causing permanent cell cycle arrest in response to various stimuli such as DNA damage, telomere dysfunction, and oncogenic activation [1,2]. Senescence can be induced via two main stimuli, intrinsic and extrinsic stimuli, through replicative and stress-induced premature senescence (SIPS), respectively. Replicative senescence is induced in response to cellular aging, usually from the DNA damage response (DDR) caused by oxidative stress, oncogene activation, radiation, mitochondrial dysfunction, and cell proliferation, leading to telomere shortening [3]. Whereas SIPS is activated in response to various extrinsic factors, such as hypoxia, the presence of adjacent senescent cells, inflammation, and immune cell activation in young and healthy cells can develop into a senescent-like phenotype with the secretion of pro-inflammatory mediators, and develops differently than replicative senescence [4]. Senescent cells undergo cell cycle arrest normally in the G1 and G2 phases, depending on the type of stimuli [5,6]. Senescence is different from quiescence (G0 phase), where cells reversibly exit the cell cycle and stop dividing; on the contrary, senescence is a permanent endpoint state, while quiescent cells are able to leave the G0 state and return to the cell cycle under the right conditions [7].

### Search Strategy and Selection Criteria

The most relevant publications included in this article were identified in the PubMed database: https://www.ncbi.nlm.nih.gov/pubmed/ (accessed on 1 January 2020) (1) Keywords used for searching (selection criteria): ischemic stroke, senescence, blood brain barrier, SASP, senolytics; neurodegeneration. (2) Dates of searching: January 2010–December 2023. 

## 2. Mechanisms of Senescence

There are two main signaling pathways activated in senescence, p53/p21(WAF1/CIP1) and p16(INK4A)/pRb [5]. Cellular senescence is a multistage process with three main stages: early, full, and late senescence [6,8]. Early senescence is marked by the activation of p53/p21 and p16 pathways in direct response to stimuli [9]. Compared to the other two-pathway activation that induces senescence, the full senescence development stage includes chromatin remodeling via nuclear lamin B1 (LB1), where the loss of LB1 reduces cellular proliferation, changes the transcriptional profile and is considered as a biomarker of senescence [10,11]. This change in transcriptional profile affects the major senescence programs, called senescence-associated secretory phenotype (SASP), SA beta-galactosidase, and mitochondrial metabolism [6,10,12]. This is normally the end of the senescent stages but, depending on the intensity of the stimuli as well as the duration of the senescent state, the cell will enter into late senescence [6,13]. In late senescence, cells undergo phenotypic diversification and differential expression of senescence markers depending on cell type, and change the composition of SASP secretion [6]. SASP includes various pro-inflammatory mediators, chemokines, extra-cellular matrix remodeling factors, growth factors and non-protein components, such as reactive oxygen species (ROS) and nitric oxide (NOS) [14]. 

Expanding the role of the p53/p21 and p16 pathways allows for a better understanding of senescence, and subsequently SASP activation. p53 is activated in response to DNA damage caused by telomere attrition, oxidative stress, or oncogenic stress [15]. Additionally, constitutive DNA damage response (DDR) signaling leads to chronic activation of p53, which induces cellular senescence [16]. Activation of p53 induces the activation of anti-proliferative genes to bring the cell to exit the cell-cycle and stop dividing [17]. This activation is also dependent on several modifying factors, including phosphorylation, ubiquitination, neddylation, sumolyation, methylation, and acetylation [5,18]. In addition, P21 is activated by p53, and they act together to inactivate all cyclin-dependent kinases (CDKs), the modulators of progression in the cell cycle. Inhibition of the cell cycle progression is through the phosphorylation of the retinoblastoma protein (RB) family, resulting in association and binding to E2Fs and subsequent formation of the DREAM (DP, RB, ETF4, and MuvB) complex that suppresses all cell cycle genes [5,19,20]. P21 itself is vital for the initiation of senescence, whereas p53 expression without p21 would induce cellular quiescence rather than senescence [21,22]. While p21 is crucial for induction of the senescent program, it does not persist in senescent cells, with continued expression of p53 and p16 required for senescent maintenance [5,23,24]. 

Compared to p53/p21 signaling, the p16 pathway controls continued senescence and has distinct stimuli for senescent induction. The RB family of proteins is one of the main targets for cyclin–CDK complexes and bind to E2F complexes, a process which then inhibits the transcription of genes required for cell cycle progression [25,26]. In addition, the RB protein interaction with the mitogenic AKT signaling pathway plays a key role in transforming quiescent cells into a senescent cell state through regulation of FOXO3 and FOXM1 [5,27,28]. The Ink4/ARF locus encodes for three tumor suppressors; p16INK4A, p14ARF encoded by *CDKN2A* gene, and p15INK4B by *CDKN2B* gene, which are all responsible for cell cycle progression control: p16 and p15 by binding to CDK4/6 and p14 by regulating p53 [29,30]. Continued senescent expression in cells is, therefore, dependent on p16 after induction of senescent programs via p21 and p53. Additionally, compared to p21, p16 expression can be induced via epigenetic stimuli, while p21 expression is induced by intrinsic DDR stimuli [29,31]. 

## 3. Senescence Associated Secretory Phenotype (SASP)

SASP occurs in all senescent cells and is generated by a multitude of different pathways, cellular insult, DNA damage, oncogenic transformation, inflammation, and oxidative stress, and forms a pro-inflammatory phenotype that releases deleterious cytokines and other markers to the surrounding cellular environment through paracrine signaling [8,32,33,34]. This release of pro-inflammatory cytokines and other senescent markers is the basis for many age-related diseases [35]. As an individual ages, they become more susceptible to diseases and insults and this frailty is a major aspect of SASP associated inflammation [36]. The pro-inflammatory cytokines released from senescent cells spread to neighboring cells, inducing frailty cascades, which increases susceptibility to adverse health outcomes [36,37]. The presence of circulating cytokines is a crucial trait for frailty in aged individuals and the paracrine signaling mechanisms associated with SASP further supports this issue, lowering the protection of aged individuals [8,36].

Inflammation plays a vital role in SASP and during aging. Pro-inflammatory cytokines and other immune cells induce low levels of sterile chronic inflammation in a process called inflammaging [35,38]. Many of the systems which induce inflammaging, e.g., IL-6, IL-1a, TNFα, and NLRP3, are upstream components of SASP and their expression causes increased aging pathologies and inflammation, furthering the impact of SASP on neighboring cells [35,39,40]. In addition to chronic low levels of sterile inflammation, active inflammation from a harmful insult to the cell and natural aging affect the NF-kB inflammatory pathways involved with SASP and inflammaging. NF-kB is a major upstream regulator of pro-inflammatory cytokines, chemokines, and signaling mechanisms, such as mTOR, MAPK, and protein kinase B [35]. Activation of these pathways by secondary SASP and inflammatory markers causes a pro-inflammatory feedback loop of autocrine and paracrine senescence that in turn causes vascular degeneration and immune disorders [35]. Additionally, another aspect of circulating pro-inflammatory cytokines inducing chronic inflammation is the reprogramming of the hematopoietic-vascular niche. This reprograming due to aging suppresses the restorative mechanics of the body for the accurate repair of injured organs and cells, and results in a higher level of chronic inflammation by circulating pro-inflammatory cytokines like IL-1α [41]. For example, endothelial cells become reprogrammed and contribute to platelet activation, which further impedes regenerative capacity and more easily induces a fibrotic phenotype, which can play a major role in the severity of an ischemic lesion [34,41].

## 4. Mechanisms of SASP Activation

Senescent markers of SASP and phenotypic changes occur during full senescence, when the cell has undergone transcriptional remodeling and is fully exited from the cell cycle [2,6]. SASP secretes many different growth factors, cytokines, and proteases into the extra-cellular environment, with a few, such as IL-1α and IL-6, performing both an autocrine pro-senescent feedback loop and paracrine signaling to adjacent cells [2,33,42]. In addition, SASP expression is modulated differently depending on the type of senescent induction, length in senescence, cell type, and location [5,42,43]. Within all these differences, the most commonly conserved mechanism is the NF-kB signaling pathway, inducing the pro-inflammatory factors IL-6 and IL-8 that are central components of SASP and are heavily expressed [1,8,44]. To that extent, DDR is the most common cause of SASP expression, with the specific stimuli showing different severities and extent of the SASP [5,45,46]. NF-kB is so highly conserved that there is another SASP regulation pathway via p38MAPK that induces NF-kB activity independent of DDR [45]. 

Aside from SASP factors inducing a positive autocrine feedback loop in senescent cells, they can also induce and reinforce senescence and SASP in neighboring cells in their environment with paracrine signaling. Not all SASP factors contain positive autocrine feedback loops, like IL-1α and IL-6, to continue senescence, but contain paracrine factors [47]. These factors can be secreted into the extra-cellular environment in a paracrine fashion to change non-senescent healthy and dividing cells into senescent cells by inducing DDR and ROS signaling [48,49,50,51]. SASP itself has many different temporal and spatial stages that can be affected in several ways but can be broken into three major timescales: initially with a rapid DDR-associated phase, then a slower self-amplification phase, which later develops into the late or “mature” phase of SASP [52]. 

There are many different factors which regulate and promote SASP both in the nucleus and cytoplasmic regions of senescent cells. Factors like mTOR, p38MAPK, JAK2/STAT3, inflammasome, and ATM, result in downstream activation of the NF-kB and CEBP-beta transcription factor pathways [2,35,44,45,53,54,55].

The upstream regulator of inflammatory senescence and SASP is the mammalian target of rapamycin (mTOR), producing IL-6 and IL-8 via IL-1a and NF-kB expression [1,2,34]. IL-1α plays a major role in induction and continuation of SASP when modulated by mTOR following an inflammatory insult [1,44].

Aging-associated SASP occurs due to multiple factors; targeting the main upstream pro-inflammatory regulator, IL-1α, one can better understand the cross interactions of the pro-inflammatory signaling cascade between stroke and SASP. Cell surface bound IL-1α is essential for pro-inflammatory SASP and for the subsequent pro-inflammatory secretion of IL-6 and IL-8 [1]. Canonically, IL-1α is not cleaved until severe injury to the cell or cell death, which occurs in the ischemic core and penumbra after stroke. As a result, noncanonical IL-1α cleavage, which allows maturation while the cell is still alive, is crucial for non-interrupted SASP activation in senescent cells with aging [2,56]. Therefore, unlike canonical cleavage by caspase-1, IL-α in senescent cells is cleaved by caspase-5/11, letting IL-1α act in both a paracrine and autocrine fashion to induce IL-1α dependent SASP [56]. Upon cleavage, IL-1α binds to its receptor IL-1R, forming a complex with both its specific and co-receptor (IL-1RAcP), which then activates NF-κB that transactivates genes for IL-6 and IL-8 [1,57]. This induces a pro-inflammatory signaling cascade, which both increases circulating pro-inflammatory cytokines and also induces other cells to adapt to a senescent phenotype [1,33,56]. 

The increase of basal pro-inflammatory cytokine expression is due, in part, to the aging of microglia, one of the main sources of IL-1α in the brain, and subsequent activation [58,59]. Normal activation of microglia, which secretes pro-inflammatory cytokines, only occurs when a threat is present, and then anti-inflammatory cytokines are released to return the system to homeostasis. As microglia age, there is a lack of this homeostatic return due to telomeric damage and inflammation, resulting in chronic activation of microglia, and leading to pro-inflammatory paracrine signaling cascades for the surrounding cells, especially in the neurovascular niche, causing secondary senescence due to SASP upregulation [58,60]. Increase of IL-6 from SASP positive cells has been observed in chronically activated brains, inducing more IL-1α mediated SASP, furthering the cycle and inducing neurotoxic astrocytic phenotypes, that directly harm and hamper regenerative and protective effects against insult [58,59,60].

## 5. Role of Senescence/SASP in Ischemic Stroke

Ischemic stroke is induced by occlusion of blood vessels supplying nutrients and oxygen to the brain regions, resulting in a loss of tissue homeostasis and neurological function. Ischemic reperfusion (IR) causes a series of cellular events, such as oxidative stress and neuroinflammation, which ultimately culminate in cellular damage. Senescence is an intrinsic feature of aging and aging is one of the major risk factors (and the biggest) for ischemic stroke [61,62]. Accumulating evidence has demonstrated the contribution of senescence in the pathogenesis of various neurovascular diseases [63,64]. Acute brain injury, such as traumatic brain injury (TBI), also causes SASP transcriptional profile in a mouse model of controlled cortical impact (CCI) [65]. However, the contribution of senescence and SASP factors in ischemic brain injury within the insult area or the cellular phenotype in this biological context is poorly described. The contribution of senescence and SAP is depicted in Figure 1.

Ischemic injury-mediated stress-induced premature senescence (SIPS) is an exceptionally complex process and has not been fully revealed. IR injury induces reactive oxygen species (ROS) that cause oxidative stress and mitochondrial damage, such as mitochondrial autophagy, that in turn causes senescence through the activation of p21 and p16 pathways [66]. Increased ROS and mitochondrial damage are considered as damage-associated molecular patterns (DAMPs), which cause extravasation of neutrophils and excessive release of pro-inflammatory cytokines, which contributes to SIPS [67]. Conversely, senescent cells induce the senescence of adjacent cells by secretion of SASP, further amplifying the inflammatory response [66]. In addition, experimental evidence suggests that epigenetic modifications and activation of the p53/p21 and p16/pRb pathways [9] are involved in cellular senescence in IR injury [68]. Epigenetic modification is defined as the modification of gene expression without altering DNA sequence through acetylation, methylation, LncRNA, and miRNA [69,70,71,72]. Elevated H3 histone acetylation caused G2/M cell cycle arrest and p53 acetylation induced premature senescence in renal epithelial cells [9,68]. Similarly, IR injury induces continual DNA damage response and induces a cell cycle arrest signaling cascade by activating the p53/p21 and p16/pRb pathways [9,73]. However, all of these current studies were conducted in peripheral tissues such as the kidney and heart; the mechanism of senescence and SASP activation has not been explored in ischemic stroke.

It is striking that many senescence-inducing stresses are actively involved in the patho-physiology of ischemic stroke. Recently, the involvement of cellular senescence and SASP in brain injury and neurological degeneration has been increasingly recognized [74]. Experimental evidence suggests that ischemic stroke triggers SASP, thereby releasing cellular factors, such as *p16*, *p21*, *Il6*, *Tnfa*, *Cxcl1* and *Cdk4*, and chemokine receptor *Cxcr-2* mRNA in the ipsilateral region penumbra as early at 30 min post middle cerebral artery occlusion (MCAO), and robust activation was observed 72 h post-MCAO [33]. Among these, *p16*, *p21*, *Cxcl*, *Cxcr2* and *Tgfb2* are expressed predominantly in the infarct region, where as *Il6* and *Tnfa* are expressed mainly in the contra-lateral area. These markers are primarily localized in the neurons and microglia with membranous and cytoplasmic localization as DNA damage response (DDR). Neuronal senescence has been implicated in the pathogenesis of various neurodegenerative diseases [75]. Increased SASP activation is associated with stimulation of the p65/NF-κB pro-inflammatory pathway and increased expression of DNA reparation foci (γH2AX) in microglia. NF-κB is a master regulator of SASP transcriptional activation [76]. Senescence was initially observed in the penumbra and later in the ischemic core, and elevated *p21* levels may protect the neuron by arresting the cell cycle. Senescence-associated beta-galactosidase (SA-β-gal) is the marker for senescence, showing lack of SA-β-gal activity, a common senescent marker, in the infarct brain at 72h post MCAO, indicating later onset of senescence. Immuno-histochemical analysis of an autopsy human brain sample showed that an increased p16 level was observed in the perimeter of the infarct area [33]. Similarly, increased mRNA expression of *p16*, *Il-6*, *Ccl8*, and *Cxcl2* were detected in the ipsilateral cortex, striatum, and hippocampus from 6-12h in a mouse model of MCAO [77]. Immuno-histochemical analysis revealed that increased expression of senescent markers was predominantly localized in astrocytes and endothelial cells rather than neurons and microglia in the ipsilateral region. The percentage of senescent neurons, microglia, astrocytes, and endothelial cells increased in aged mice compared to young mice, demonstrating that acute senescence-induced brain pathologies closely resemble the aged brain. Cells enter senescence in response to various cellular injuries, characterized by increased resistance to apoptosis, cell cycle arrest, and increased SASP [78]. In a rat model of ischemic stroke, cellular senescence was evident at 7-days post-stroke, indicated by lipofuscin accumulation in the cortical and caudate-putamen areas of ischemic infarct, increased cell cycle arrest mediators, such as p21, p53 and p16, and increased SASP secretion [79]. Lipofuscins are non-degradable lysosomal complexes composed of proteins, sugars, ions and lipids, especially in post-mitotic cells. As it accumulates in the lysosome of the aged postmitotic cell, it is considered as a lysosomal marker of senescence, along with SA-β-gal [80]. Increased accumulation of lipofuscin was detected earliest at 24-h post MCAO, suggesting a targetable pathogenic incident. In contrast to previous study by Torres-Querol et al. [33], Baixauli-Martin et al. [79] proposed lipofuscin as a superior marker of senescence over SA-β-gal. In silico analysis of bulk RNA sequence data from public databases showed a differential expression of senescence-related genes, such as angiopoietin-like protein (ANGPTL4), *Ccl3*, *Ccl7*, *Cxcl16*, and *Tnf*, after stroke in a species-conserved manner [81]. ANGPTL4 is an endothelial protein associated with the pathogenesis of ischemic stroke and inflammation [82]. The expression of these genes was further confirmed by qPCR analysis in rat brain tissues after MCAO. A Further single RNA sequence demonstrated that senescence-related genes are highly expressed in microglia and monocytes, more precisely in MG4 microglia, a sub-population of microglia involved in blood–brain barrier integrity [81]. A summary of studies on the role of senescence and SASP in ischemic stroke is listed in Table 1. All together, these studies highlight the involvement of senescence and SASP in the pathogenesis of ischemic stroke and suggest it as a potential therapeutic target.

The impact of stroke morbidity and mortality increases with age, therefore understanding the implications of aging, through cellular senescence, is vital. Currently, only a single study has evaluated the effect of modulation of senescence and SASP on the functional outcome of ischemic stroke, ref. [83] showing that ABT263 treatment improved post-stroke neurological functions. While current literature does not fully define the functional implications of senescence in stroke, there are still many inferences that can be made regarding other CNS pathologies. Stroke is a vascular disease and vascular senescence causes a myriad of downstream effects on the integrity of the BBB. BBB permeability is an early feature of age-related diseases, including Alzheimer’s disease, due to vascular senescence [74,84]. Chronic inflammation induced by SASP activation increases BBB permeability [85], reduces neurogenesis, and adversely affects cognitive performance [86]. Previously, it has been shown that clearance of senescent astrocytes and microglia improved working memory in a mice model of Alzheimer’s disease [87]. Consistently, clearance of senescent cells in the brain through genetic and pharmacological approaches had beneficial consequences for aging-associated cognitive function [88]. Based on these finding, we anticipate that modulation of cellular senescence may alleviate post-stroke cognitive impairment.

## 6. Senescence/SASP-Targeted Therapeutics for Ischemic Stroke

Clearence of senescent cells is considered as a therapeutic approach to mitigate neurofibrillary tangle deposition, gliosis, and neuronal degeneration, thereby preserving cognitive function in a mouse model of Alzheimer’s disease [87]. Recently, senolytic drugs have gained scientific attention as they specifically target anti-apoptotic proteins induced by senescence, selectively eliminate senescent cells [89] and have been validated in animal models of various neurodegenerative diseases [87,90], leading to innovative cures for age-related ailments. ABT263 (navitoclax) and BH3-mimetic [91] selectively eliminated senescent astrocytes following oxygen-glucose deprivation in rat cortical astrocytes by inducing apoptosis [83]. ABT-263 is an anti-cancer drug that selectively targets and disrupts the interaction between Bcl2 and Bcl-xL, thereby inducing the apoptosis of senescent cells [92]. ABT-263 has shown its beneficial effect on neuronal damage induced by Cisplatin, a chemotherapy medication, and by whole-brain irradiation [93,94]. Further, ABT-263 attenuated reactive astrogliosis, inducible nitric oxide synthase-2 (NOS-2) expression and neutrophil activation in a rat model of cerebral IR injury [83]. NOS-2 is activated by ischemic stroke, which leads to oxidative stress, inflammation and necrosis of infarcted tissue [95]. NOS-2 is also considered as a marker for microglial activation [96]. In addition, senolytic treatment reduced infarct volume and improved neurological functional outcomes. ABT-263 improved motor and neurological function and modulated endothelial and astrocytic P16 expression. Although ABT-263 has shown neuroprotection in mouse models of ischemic stroke, this beneficial effect may have originated from generalized systemic off-target effects, such as such as thrombocytopenia, produced by the drug [97,98], which can damage neighboring non-senescent cells [99]. To counteract any later non-specific effect, Lu et al. demonstrated that the ipsilateral administration of a novel lenti-INK-ATTAC viral vector that delivers ATTAC (apoptosis through targeted activation of caspase) gene, which was specifically designated to eliminate senescent cells in the brain, could provide neuroprotection in ischemic stroke. INK-ATTAC causes the inducible elimination of p16 (Ink4a)-positive senescent cells upon administration of the inducer, AP20187 [77], demonstrating that local clearance of senescent cells is a possible therapeutic avenue for ischemic stroke. 

## 7. Conclusions

Based on these observations, senescence and SASP play a critical role in ischemic stroke and elimination of these senescent cells could be a possible therapeutic approach. Seno-therapeutics have been developed to prevent the harmful effect of senescence and SASP, and include synolytics, senomorphs, rejuvenating agents and stem cell-based therapies [73]. Previous studies demonstrated that ABT-263, FOXO4-D-Retro-Inverso peptide, and a combination of dasatinib and quercetin modulated inflammatory response and eliminated senescent cells, thereby attenuating IR associated cardiac and renal injury [100,101,102,103]. However, ABT-263 has only been validated in animal model of ischemic stroke [77,83]. Senomorphs are agents that prevent cell cycle arrest, rather than eliminating senescent cells, and attenuate the secretion of SASP [104]. Metformin, an mTOR inhibitor, and rapamycin, an AMPK activator, are the most commonly studied senomorphics and have been reported to attenuate IR-associated renal and cardiac injury and SASP secretion [105,106]. Rejuvenating agents, such as resveratrol, are known to protect from IR injury by activating SIRT1 [107]. SIRT1 alleviates senescence by activating cell division and inhibits senescence by deacetylation of p53. In addition, mesenchymal stem cell (MSC) therapy is another therapeutic avenue for alleviating the premature senescence associated with IR injury [107,108,109]. MSCs secret various soluble factors and exosomes with immunomodulatory properties, which can inhibit SASP and attenuate senescence [108,110,111]. Recently, Xiao et al. demonstrated that MSC-derived extra-cellular vesicles stimulate angiogenesis and attenuate endothelial senescence [112]. In general, further validation of these approaches is needed in the context of ischemic stroke to explore the clinical application of such interventions.

Currently, our understanding of the mechanisms of brain senescence and SASP activation in the context of ischemic stroke is at the tip of the iceberg, which warrants further analysis. Cellular senescence is a complex dynamic event, and the lack of specific biomarkers limits its further investigation. Recent advances in spatial and single cell transcriptomic approaches present a viable option to characterize cellular complexity, along with identification of triggers and unique regulatory molecules, and impact on healthy cells. Research in this area is ongoing, and understanding the interplay between senescence and stroke is crucial for developing targeted therapies to enhance recovery. Experimental evidence of senescence and SASP activation from peripheral organs, such as heart and kidney, have laid the foundations for brain researchers to explore the basic mechanism of senescence and SASP activation in ischemic stroke and to develop novel therapeutic approaches.

## Figures and Tables

**Figure 1 pharmaceuticals-17-00312-f001:**
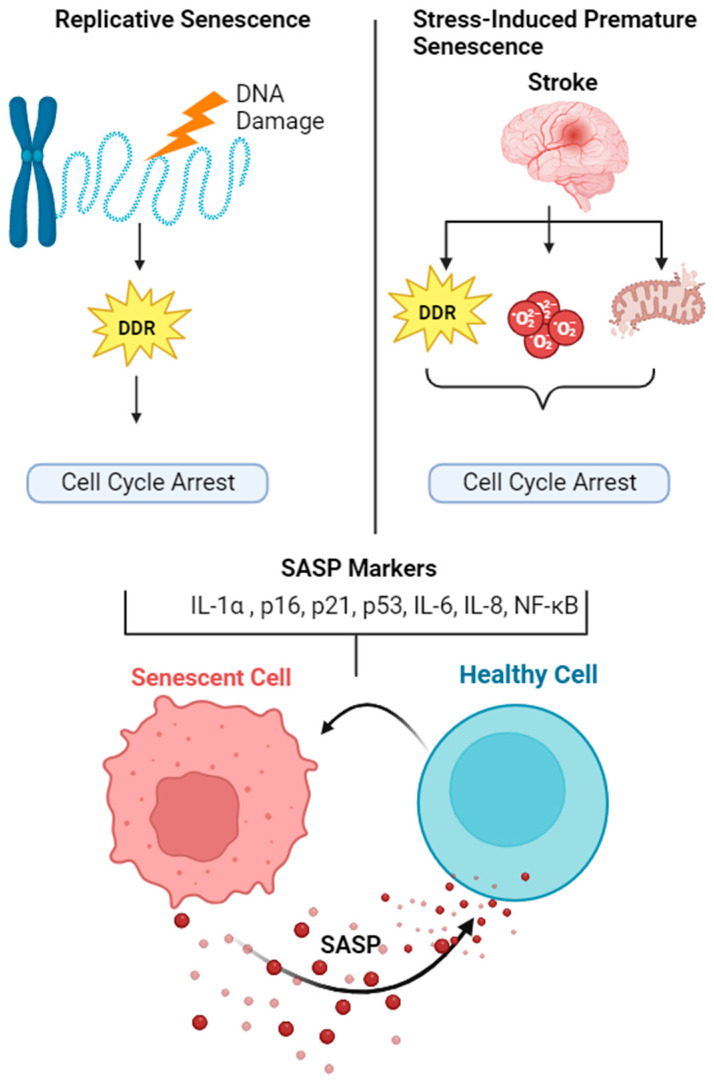
The contribution of senescence and SASP in aging and stroke. Senescence is an integral component of aging and aging is an independent risk factor for ischemic stroke. Aging causes replicative senescence due to persistent telomere shortening, whereas stroke causes stress induced premature senescence (SIPS). Although senescent cells undergo cell cycle arrest, they are metabolically active and secrete various growth factors, inflammatory mediators and extra-cellular matrix modifying components, termed as senescent associated secretory phenotype (SASP). SASP is contagious in nature, stimulating adjacent non-senescent cells to undergo senescence and secrete SASP. Hence, senolytic therapies are proposed to selectively eliminate SASP and senescent cells in order to counteract the detrimental effect of ischemic stroke.

**Table 1 pharmaceuticals-17-00312-t001:** Summary of studies on contribution of senescence and SASP in ischemic stroke.

Animal Model	Treatment	Inference	Reference
MCAO in CD1 male mice	Nil	Increased senescence markers such as p16 and p21 in the infarct area with neuronal and microglial localization. Increased pro-inflammatory cytokines (Il6, Cxcl1, Tnfa and Cxcr2) along with P65 NF-κB and γ-H2AX in the brain of tMCAO in mice 72 h after the ischemic stroke.	Querol et al. 2021 [33]
Male Sprague–Dawley (SD) rats model of transient MCAO	ABT263 (10 mg/kg) for 3 days	ABT263 reduced the infarct volume and improved neurological outcomes following MCAO ABT263 treatment attenuated expression of NOS2, neutrophil activation and SASP induced MCAO	Lim et al. 2021 [83]
Wistar rat model transient MCAO	NIL	MCAO-induced senescence indicated lipofuscin accumulation (7 days post-stroke), increased mRNA expression of the Cdkn1a/p21, Tp53, and Cdkn2a/p16 and SASP cytokines Il6, Tnfa, and Il1b	Martin et al. 2022 [79]
C57BL/6 male mice model transient MCAO	1. ABT-263 administrated 24 h post-MCAO (50 mg/kg/day) for 5 days 2. stereotaxical injection of Lenti-INK-ATTAC	Acute SASP activation(p16INK4a, IL-6, CCL8, and CXCL2) in the ipsilateral side of the mice after MCAO ABT-263 improved neurological outcome and eliminated mRNA of p16^INK4a^ in endothelial cells and astrocytes at 7 days after MCAO Lenti-INK-ATTAC infection attenuated SASP and expression of p16^INK4a^ in the brain tissue of MCAO mice	Lu et al. 2023 [77]
Rat model of permanent MCAO	Nil	Increased ANGPTL4, CCL3, CCL7, CXCL16, and TNF after stroke	Fu et al. 2023 [83]

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
