# Peer review of "Senescence and SASP Are Potential Therapeutic Targets for Ischemic Stroke"

_pharmaceuticals, 2024, doi:10.3390/ph17030312_

Round 1

Reviewer 1 Report (New Reviewer)

Comments and Suggestions for Authors

This is a well-written review of post-' stroke ischemic brain damage.  It comes from a well respected institution with good research ongoing in stroke management. Strengths of the paper include the ability of the authors in judging the relevance of published information in the research field. I do not see any major research weaknesses. One small issue is the spelling to the cancer treatment agent cisplatinum as cisplan.

Comments on the Quality of English Language

I think this paper is well written and is suitable as a review for publication. I see no major language  issues except the spelling of cisplatin.

Author Response

This is a well-written review of post-' stroke ischemic brain damage.  It comes from a well respected institution with good research ongoing in stroke management. Strengths of the paper include the ability of the authors in judging the relevance of published information in the research field. I do not see any major research weaknesses. One small issue is the spelling to the cancer treatment agent cisplatinum as cisplan.

Response: Thank you for your thoughtful and encouraging comments on my work. We deeply appreciate the time and effort you invested in providing such insightful remarks.

Cispan is corrected as suggested.

Reviewer 2 Report (New Reviewer)

Comments and Suggestions for Authors

In this review entitled: 'Senescence and SASP are Potential Therapeutic Targets for Ischemic Stroke', the authors examined the stroke-dependent mechanisms of senescence through the activation of p21 and p16 pathways leading to senescence-associated secretory phenotype. Notably, therapies that target the senescent cells and SASP are not well described in the literature, so any insights into the role of senescence mechanisms and its potential therapeutic targets are very important.

Overall, this comprehensive review is well-researched, and the authors have meticulously compiled information linking major molecules/signaling pathways in senescence and stroke and illustrated the contribution of senescence and SASP in aging and stroke.

I have minor criticisms for this work:

Authors should decipher the abbreviations:

ATM - page4, 

SA-B-gal -page6

Author Response

In this review entitled: 'Senescence and SASP are Potential Therapeutic Targets for Ischemic Stroke', the authors examined the stroke-dependent mechanisms of senescence through the activation of p21 and p16 pathways leading to senescence-associated secretory phenotype. Notably, therapies that target the senescent cells and SASP are not well described in the literature, so any insights into the role of senescence mechanisms and its potential therapeutic targets are very important.

Overall, this comprehensive review is well-researched, and the authors have meticulously compiled information linking major molecules/signaling pathways in senescence and stroke and illustrated the contribution of senescence and SASP in aging and stroke.

Response: We deeply appreciate the time and effort you invested in providing such insightful remarks.

I have minor criticisms for this work:

Authors should decipher the abbreviations:

ATM - page4, 

Response: Updated as suggested.

SA-B-gal -page6

Response: Updated as suggested.

Reviewer 3 Report (New Reviewer)

Comments and Suggestions for Authors

I appreciate the opportunity provided by the editor to review this intriguing manuscript. The current review investigates the involvement of cellular senescence in ischemic stroke. The manuscript is well-written and scientifically sound. I have a few comments:

1. I strongly recommend that the authors include a methodological section on the conduct of the literature search following the PRISMA updated guidance (Liberati et al., 2009; Page et al., 2021) and include a flow chart.

2. I suggest dedicating a paragraph to a more in-depth explanation of the functional implications of senescence in ischemic stroke. Expanding on how senescence may be linked to medium to long-term functional outcomes would provide valuable insights for readers.

3. Authors report in Table 1 the studies investigating senescence in IS and related animal models. However, there is no actual discussion of those papers.

4. I suggest including a separate paragraph discussing the therapeutic perspectives. This would enhance paper readability and align with the main aim of the review.

Minor:

On page 6, there are two periods at the end of a sentence.

Comments on the Quality of English Language

The manuscript is well-written

Author Response

I appreciate the opportunity provided by the editor to review this intriguing manuscript. The current review investigates the involvement of cellular senescence in ischemic stroke. The manuscript is well-written and scientifically sound. I have a few comments:

Response: Thank you for reviewing the manuscript. We are grateful to receive such thoughtful comments and suggestions on areas in which our manuscript can be improved.

  1. I strongly recommend that the authors include a methodological section on the conduct of the literature search following the PRISMA updated guidance (Liberati et al., 2009; Page et al., 2021) and include a flow chart.

Response: A paragraph is added after the introduction “Search strategy and selection criteria”. Since it is not a systematic review, we are not adding the flowchart.

  1. I suggest dedicating a paragraph to a more in-depth explanation of the functional implications of senescence in ischemic stroke. Expanding on how senescence may be linked to medium to long-term functional outcomes would provide valuable insights for readers.

Response: We added a paragraph mentioning the functional implication modulation of senescence and SAP in ischemic stroke. Although limited studies are available so far in this context, we tried to make a paragraph based on studies on various other neurovascular diseases.

  1. Authors report in Table 1 the studies investigating senescence in IS and related animal models. However, there is no actual discussion of thosepapers.

Response: We have improved the discussion of table  under the section “Role of Senescence/SASP in-targeted therapeutics for ischemic stroke”.

  1. I suggest including a separate paragraph discussing the therapeutic perspectives. This would enhance paper readability and align with the main aim of the review.

Minor:

On page 6, there are two periods at the end of a sentence.

Response: It was a typo, updated as suggested.

Round 2

Reviewer 3 Report (New Reviewer)

Comments and Suggestions for Authors

I thank the authors for their reply. I do not have further comments.

This manuscript is a resubmission of an earlier submission. The following is a list of the peer review reports and author responses from that submission.

Round 1

Reviewer 1 Report

Comments and Suggestions for Authors

The purpose of the review by Ouvrier et al was to examine the effects of senescence and SASP as a consequence of ischemic stroke in the brain. The authors start by describing cellular senescence and mechanisms of SASP activation, but give limited information regarding brain senescence after the stroke.

The review is superficial, fragmented, sometimes with no logical connection between the parts, and with multiple inaccuracies.

For example: The first sentence in Introduction reads: “Cellular senescence occurs when cells enter the biological limit in cell division…”. The second sentence contradicts to the first and shifts to a different thought: “Senescence can be induced by two main pathways, intrinsic and extrinsic, through replicative and stress induced premature senescence.“

Lines 86-89: As written, it is unclear what are unique differences between the effects of p16.

Lines 142-145: What does “ending up” mean? The mechanisms should be explained.

Lines 150-164: The authors attempt to explain the relationship between stroke and SASP and mention the contribution of aging. It is very unclear whether the authors are describing age-associated senescence or stroke-associated changes in the brain that eventually lead to senescence. This part must be rewritten.

Lines 166-175: “increase in circulating proinflammatory cytokines is due to activated microglia.” Does IL1α induced locally in microglia reach the circulation? Or does it have paracrine effects on surrounding cells? What exactly is the contribution of SASP in aging microglia? How is senescence established in microglia in normal aging? By what mechanisms?

Lines 198-207: “In addition, experimental evidence suggest that epigenetic modifications and are activated cellular senescence in IR injury” The sentence grammar should be revised and the paragraph overall is unclear. It is not necessary to explain epigenetic modification, only what kind of epigenetic modification was found in IR that is associated with senescence and SASP induction. Is it found only in peripheral tissue and the authors approximate these findings to changes in the brain?

Lines 222-227: Are the authors discussing here cellular proteins or mRNA? If it is proteins, it should be denoted by capitalizing the first letter without italics. The authors describe changes that occur 30 minutes after the experiment. These changes are acute effects of DNA damage, which may be seen even after 72 hours. Senescence develops much later. This should be discussed and acknowledged.

Lines 233-235: The authors discuss changes in several gene expression by mRNA. These genes should be denoted by capitalizing the first letter with italics. The same is true for lines 247-48, where authors discuss genes, not proteins.

The Figure is not informative without mechanistic details.

Conclusions Line 295: “..rejuvenating agents such as resveratrol are known to protect from IR injury by activating SIRT. SIRT alleviated senescence by activating cells division and inhibits senescence by deacetylation of p53.” Deacetylation of p53 results in p53 ubiquitination and an increase in proliferation. However, if cells carrying damaged DNA (i.e., senescent cells) enter the cell cycle, they usually acquire pro-oncogenic mutations. How can this be reconciled with protection in the long run?

The impression of this reviewer is that bits of information are thrown together without much understanding by the authors of mechanisms and physiological significance.

Comments on the Quality of English Language

small grammar errors

Author Response

Reviewer 1

The purpose of the review by Ouvrier et al was to examine the effects of senescence and SASP as a consequence of ischemic stroke in the brain. The authors start by describing cellular senescence and mechanisms of SASP activation, but give limited information regarding brain senescence after the stroke.

The review is superficial, fragmented, sometimes with no logical connection between the parts, and with multiple inaccuracies.

For example: The first sentence in Introduction reads: “Cellular senescence occurs when cells enter the biological limit in cell division…”. The second sentence contradicts to the first and shifts to a different thought: “Senescence can be induced by two main pathways, intrinsic and extrinsic, through replicative and stress induced premature senescence.“

Response: Thank you for the reviewer’s comment. Suggestions are updated.

Lines 86-89: As written, it is unclear what are unique differences between the effects of p16.

Response: Updated as suggested

Lines 142-145: What does “ending up” mean? The mechanisms should be explained.

Response: Updated as suggested

Lines 150-164: The authors attempt to explain the relationship between stroke and SASP and mention the contribution of aging. It is very unclear whether the authors are describing age-associated senescence or stroke-associated changes in the brain that eventually lead to senescence. This part must be rewritten.

Response: We are explaining stroke-associated senescence and it is modified as suggested.

Lines 166-175: “increase in circulating proinflammatory cytokines is due to activated microglia.” Does IL1α induced locally in microglia reach the circulation? Or does it have paracrine effects on surrounding cells? What exactly is the contribution of SASP in aging microglia? How is senescence established in microglia in normal aging? By what mechanisms?

Response: Suggestions are included now.

Lines 198-207: “In addition, experimental evidence suggest that epigenetic modifications and are activated cellular senescence in IR injury” The sentence grammar should be revised and the paragraph overall is unclear. It is not necessary to explain epigenetic modification, only what kind of epigenetic modification was found in IR that is associated with senescence and SASP induction. Is it found only in peripheral tissue and the authors approximate these findings to changes in the brain?

Response: The sentence is corrected for grammar. Although senescence and SASP are well studied in peripheral tissues, only a limited number of studies addressed brain senescence in the context of stroke. Experimental evidence of senescence and SASP activation from peripheral organs such as heart and kidney have set the foundations for brain researchers to explore the basic mechanism of senescence and SASP activation in ischemic stroke and develop novel therapeutic approaches.

Lines 222-227: Are the authors discussing here cellular proteins or mRNA? If it is proteins, it should be denoted by capitalizing the first letter without italics. The authors describe changes that occur 30 minutes after the experiment. These changes are acute effects of DNA damage, which may be seen even after 72 hours. Senescence develops much later. This should be discussed and acknowledged.

Response: We are mentioning the mRNA levels of cellular factors such as p16, p21, Il6, tnfa, Cxcl1 and Cdk4 and chemokine receptor Cxcr-2 in the ipsilateral region penumbra as early at 30 minutes and robust activation at 72 h post-MCAO. Suggestions are now updated in the text.

Lines 233-235: The authors discuss changes in several gene expression by mRNA. These genes should be denoted by capitalizing the first letter with italics. The same is true for lines 247-48, where authors discuss genes, not proteins.

Response: Updated as suggested.

The Figure is not informative without mechanistic details.

Response: A new figure is included with more mechanistic details.

Conclusions Line 295: “..rejuvenating agents such as resveratrol are known to protect from IR injury by activating SIRT. SIRT alleviated senescence by activating cells division and inhibits senescence by deacetylation of p53.” Deacetylation of p53 results in p53 ubiquitination and an increase in proliferation. However, if cells carrying damaged DNA (i.e., senescent cells) enter the cell cycle, they usually acquire pro-oncogenic mutations. How can this be reconciled with protection in the long run?

Response: We agree with the reviewer’s comment that SIRT1 activation is not always pro-survival. However multiple studies showed the neuroprotective role of SIRT1 in ischemic stroke and senescence (PMID: 37946007, PMID: 20644332). Modulation of SIRT1 with mitochondria therapy improved functional outcomes after ischemic stroke (PMID: 37946007).

The impression of this reviewer is that bits of information are thrown together without much understanding by the authors of mechanisms and physiological significance.

Response: Thank you for the comment. There are not many studies on role of senescence and SASP in ischemic stroke. We tried to include as much as we could. Still, this is an unexplored area.

Comments on the Quality of English Language: small grammar errors

Response: The manuscript is now edited for grammatical errors and typos.

Reviewer 2 Report

Comments and Suggestions for Authors

The present manuscript summarizes the current information regarding the role of cell senescencen and ischemic stroke. The review is timely and help to develop new avenue for the prevention and treatment of ischemic stroke.

Major point

1. Figure is recommended to more specifically represent and summarise the current understanding of the role of senescence in ischemic stroke. Curret figure is less informative and too abstract.

2. As for the structure of the manuscript, it is recommended to establish the section regarding the senolytic therapy or senescence/SASP-targeted therapeut for ischemic stoker, instead of mentioning in the section 5. Role of Senescence and SASP in the Pathogenesis of Ischemic Stroke. 

3. Table 1 should be cited and mentioned in the main text.

Minor points

1. English should be improved significantly. There are some syntax errors.

2. The abbreviations should be defined at their first appearance, and thereafter, abbreviations should be consistently used.

3. There are typos. Please reconfirm the accuracy of spelling.

4. line 50: Laminin is a extracellular matric protein. Presumabaly, laminin should read lamin, a intermediate filament protein lining the nuclear membrane.

5. line 53: galactose should read galactosidase.

Comments on the Quality of English Language

There are some errors in syntax and typos. English editing is recommended.

Author Response

The present manuscript summarizes the current information regarding the role of cell senescencen and ischemic stroke. The review is timely and help to develop new avenue for the prevention and treatment of ischemic stroke.

Major point

  1. Figure is recommended to more specifically represent and summarise the current understanding of the role of senescence in ischemic stroke. Current figure is less informative and too abstract.

Response: A new figure is included with more mechanistic details.

  1. As for the structure of the manuscript, it is recommended to establish the section regarding the senolytic therapy or senescence/SASP-targeted therapeut for ischemic stoker, instead of mentioning in the section 5. Role of Senescence and SASP in the Pathogenesis of Ischemic Stroke. 

Response: updated as suggested.

  1. Table 1 should be cited and mentioned in the main text.

Response: The table is now mentioned in the main text (line 282).

Minor points

  1. English should be improved significantly. There are some syntax errors.

Response: The manuscript is now edited for grammatical errors and typos.

  1. The abbreviations should be defined at their first appearance, and thereafter, abbreviations should be consistently used.

Response: Abbreviations are updated

  1. There are typos. Please reconfirm the accuracy of spelling.

Response: The manuscript is now edited for typos.

  1. line 50: Laminin is a extracellular matric protein. Presumabaly, laminin should read lamin, a intermediate filament protein lining the nuclear membrane.

Response: Corrected as suggested

  1. line 53: galactose should read galactosidase.

Response: Corrected as suggested

Comments on the Quality of English Language

There are some errors in syntax and typos. English editing is recommended.

Response: The manuscript is now edited for grammatical errors and typos.

Round 2

Reviewer 1 Report

Comments and Suggestions for Authors

This version of the review is somewhat improved, however the parts 2,3, and 4  have to be rewritten due to multiple grammar errors and inadequacy.

For example,

in Part 2:

" activation of p53 starts the process of anti-proliferative genes"..???

" CDKs, stopping cell cycle progression by inhibiting the phosphorylation of RB protein family association ..."What  does it mean inhibiting phosphorylation of association?

" with a unique difference being p16 expression inducing senescence via epigenetic stimuli compared to p21 DDR stimuli induced senescence [29, 31]..".What is a specific role of p16 in epigenetic changes?

Part 3.''..phenotype which releases proinflammatory cytokines.." Again,  what does it mean?

The last paragraph in Part 3  should be seriously edited, like it is, it is incomprehensible.

Comments on the Quality of English Language

 in many places the text is incomprehensible due to poor English 

Author Response

This version of the review is somewhat improved, however the parts 2,3, and 4  have to be rewritten due to multiple grammar errors and inadequacy.

Response: Thank you for reviewing the manuscript. We are grateful to receive such thoughtful comments and suggestions on areas in which our manuscript can be improved.

For example,

in Part 2:

" activation of p53 starts the process of anti-proliferative genes"..???

Response: The sentence is now rewritten as “Activation of p53 starts induces the process activation of anti-proliferative genes to bring the cell out ofto exit the cell-cycle and stop dividing”.

" CDKs, stopping cell cycle progression by inhibiting the phosphorylation of RB protein family association ..."What  does it mean inhibiting phosphorylation of association?

Response: The sentence is now rewritten as “In addition , P21 expression  is induced by p53, and, inactivates all cyclin-de DDR stimuli induced senescence pendent kinaeses (CDKs), the modulators of progression in the cell cycle. Inhibition of the cell cycle progression is throughthe phosphorylation of RB protein family, resulting in association and binding to E2Fs and formation of the DREAM (DP, RB, ETF4, and MuvB) complex that suppresses all cell cycle genes ”

" with a unique difference being p16 expression inducing senescence via epigenetic stimuli compared to p21 DDR stimuli induced senescence [29, 31]..".What is a specific role of p16 in epigenetic changes?

Response: The sentence is rewritten asContinued senescent expression in cells is therefore dependent on p16 after induction in senescent programs via p21 and p53. Additionally, compared to p21, p16 expression can be induced via epigenetic stimuli while p21 expression is induced by intrinsic DDR stimuli.”

Part 3.''..phenotype which releases proinflammatory cytokines.." Again,  what does it mean?

Response: The sentence is rewritten as “SASP occurs in all senescent cells and is generated by a multitude of different pathways; cellular insult, DNA damage, oncogenic transformation, inflammation, and oxidative stress; which forms an inflammatory phenotype that releases proinflammatory cytokines and other markers to the surrounding cellular environment via paracrine signaling.”

The last paragraph in Part 3  should be seriously edited, like it is, it is incomprehensible.

Response: The paragraph is now edited

Comments on the Quality of English Language

 in many places the text is incomprehensible due to poor English

Response: The manuscript is corrected for language and typos.

Reviewer 2 Report

Comments and Suggestions for Authors

The manuscript has been satisfactorily revised. There is no further comment from this reviewer.

Comments on the Quality of English Language

Although authors responded to the comment on languge saying that they editted the manuscript for grammatical errors and typos, it is recommeded for the editorial office to give the English editting to the manuscript for the final ckeck.

Author Response

The manuscript has been satisfactorily revised. There is no further comment from this reviewer. Although authors responded to the comment on languge saying that they editted the manuscript for grammatical errors and typos, it is recommeded for the editorial office to give the English editting to the manuscript for the final ckeck.

Response: Thank you for reviewing the manuscript. We are grateful to receive such thoughtful comments and suggestions on areas in which our manuscript can be improved. The manuscript is now edited for grammar and typos

Round 3

Reviewer 1 Report

Comments and Suggestions for Authors

The manuscript has not been reviewed and edited properly, the parts 1-4 are still incomprehensible. 

Comments on the Quality of English Language

English was not improved since the last review